# The Structure of Micro-Variability in the WEBT BL Lacertae Observation

**James R. Webb** * and **Ivan Parra Sanz**

Physics CASE Faculty, Florida International University, Miami, FL 33199, USA; iparr011@fiu.edu
*   Correspondence: webbj@fiu.edu

**Abstract:** We present the results of an in-depth analysis of the Whole Earth Blazar Telescope (WEBT) micro-variability observations made during a campaign done in 2020 on the blazar BL Lacertae. The data consisted of 231 days of optical imaging and we separated the long-term light curve into individual single-night light curves, and then chose 41 nights that contained over 100 individual observations and also showed micro-variations well above the noise. Micro-variability is defined as excursions in the order of 0.01–0.1 magnitudes over timescales of hours or minutes either above or below a linear background sampled over the entire night. We then fit each individual micro-variability curve with model pulses from turbulent cells using the turbulent jet model.. We present the results of the pulse fitting analysis, which yields turbulent cell sizes, amplitudes and turbulent plasma characteristics.

**Keywords:** blazars; quasars; micro-variability; optical variability

## 1. Introduction

We use data collected by the WEBT campaign in 2020 [1] to test the turbulent jet model [2,3]. The concept of turbulence in blazar jets was first proposed by [4] to explain the lack of polarization seen in blazar radio jets, and models built on this assumption are successful in explaining the multi-wave variability of blazars [5]. Theoretical analysis and simulations of relativistic jets [6,7] show the properties found in non-relativistic turbulent jets are similar to the properties in a relativistic flow. The model presented here assumes that micro-variations are caused by turbulence in the jet flow.

Turbulence is a stochastic process, so each micro-variability curve is a realization of that stochastic process. The Kolmogorov scale in turbulence is associated with the smallest cell sizes and locations where most of the dissipation takes place in non-relativistic plasma. The largest length scales can be associated with the width of the plasma jet or the correlation length of the plasma. Turbulent relativistic extragalactic jet simulations exhibit a similar relationship between vortex length scale and energy as those in non-relativistic simulations [7]. When the shock wave encounters a turbulent cell of a specific size, density and magnetic field orientation, this results in particle acceleration, and a pulse of synchrotron radiation is emitted as the shock passes through the cell. Each micro-variability light curve is a realization of the underlying turbulent plasma jet. De-convolving each of the light curves into pulses allows us to estimate the sizes and numbers of the turbulent cells in the jet. The observations are described in Section 2, the model details are presented in Section 3 and the results of fitting the model pulses to the light curves are presented in Section 4.

## 2. Observations

BL Lac is a popular target for blazar observers because it is a very bright and active source which has shown micro-variations with a duty cycle of 0.54% [2]. The WEBT collaboration observed BL Lacertae over a period of 231 nights during the 2020 campaign.

During that period, WEBT telescopes around the world collaborated to make a nearly continuous light curve. Figure 1 shows the entire WEBT R-band curve (blue) along with unpublished Florida International University FIU data (tan). The abscissa is modified Julian date minus 2450000.5. The data are not continuous; there are daily gaps.

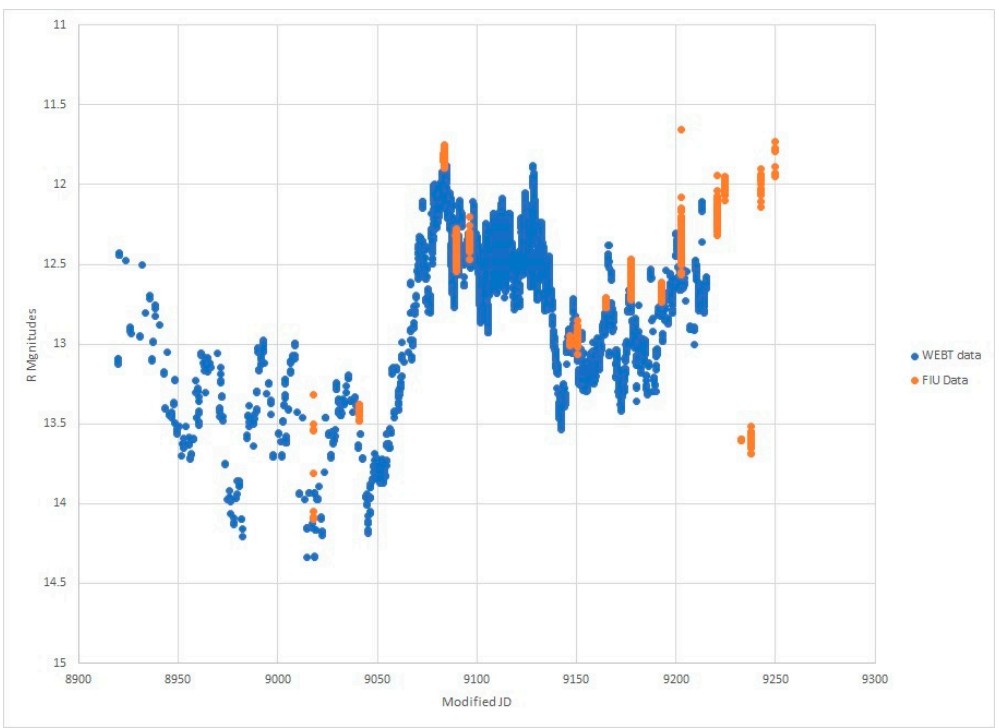

**Figure 1.** WEBT light curve of BL Lac with FIU data superimposed.

We separated the light curve up into individual night light curves and concentrated on 41 well-sampled light curves, with each night having over 100 individual images taken and BL Lac showing micro-variations during that night.

## 3. Data Analysis

### 3.1. Theory of Shock Waves Encountering Turbulent Cells

We applied the model presented in [2,3], in which micro-variations are modeled as a convolution of pulses of synchrotron radiation emitted as a shock propagating down the relativistic jet. We interpreted the blazar micro-variability light curves as pulses or shots since they do not show the characteristics of random noise or full-wave periodic oscillations. Many measurements of the rise time and decay time of trends in the micro-variations tended to yield consistent values in many different light curves and many different sources, indicating some type of structure was present. The goal became to find a reasonable model to explain the consistently shaped pulses in the blazar jet environment. Lehto [8] first investigated using shot models to describe the light curve power spectra seen in the X-ray light curves of active galaxies. The Lehto model was based on a delta function rise and exponential decline and a random distribution of such shots, which can yield a 1/f power spectrum. There was no physical basis for this pulse model and the micro-variability pulses seen in the blazar light curves did not resemble the Lehto pulses. However, the idea that the micro-variations could be de-convolved into individual pulses was an attractive interpretation. The analysis of hundreds of Blazar micro-variability light curves from the Florida International/SARA blazar monitoring program and from other observers [9] led [2] to propose that the micro-variability curves exhibited shot profiles similar to the profiles derived by [6]. The model predicts that a shock wave encountering a cylindrical density enhancement in the plasma accelerates the electrons, which then cool by

emitting synchrotron radiation. Individual turbulent cells being energized by a plane shock propagating down the jet as computed following [6] (hereafter KRM) became the basis of our model. The individual synchrotron pulses are the result of the shock encountering individual turbulent cells with a range of sizes, densities, magnetic field strengths and magnetic field orientations.

The burst profiles described by the KRM model matched the micro-variability pulse profiles seen in the actual blazar light curves. The flux rise and decline of the model pulse closely matched the observed blazar micro-variability pulses. Our model involves convolving many KRM pulses together since the shock wave would encounter multiple turbulent cells as it propagated down the three-dimensional jet. The profile for a single KRM pulse is shown in Figure 2.

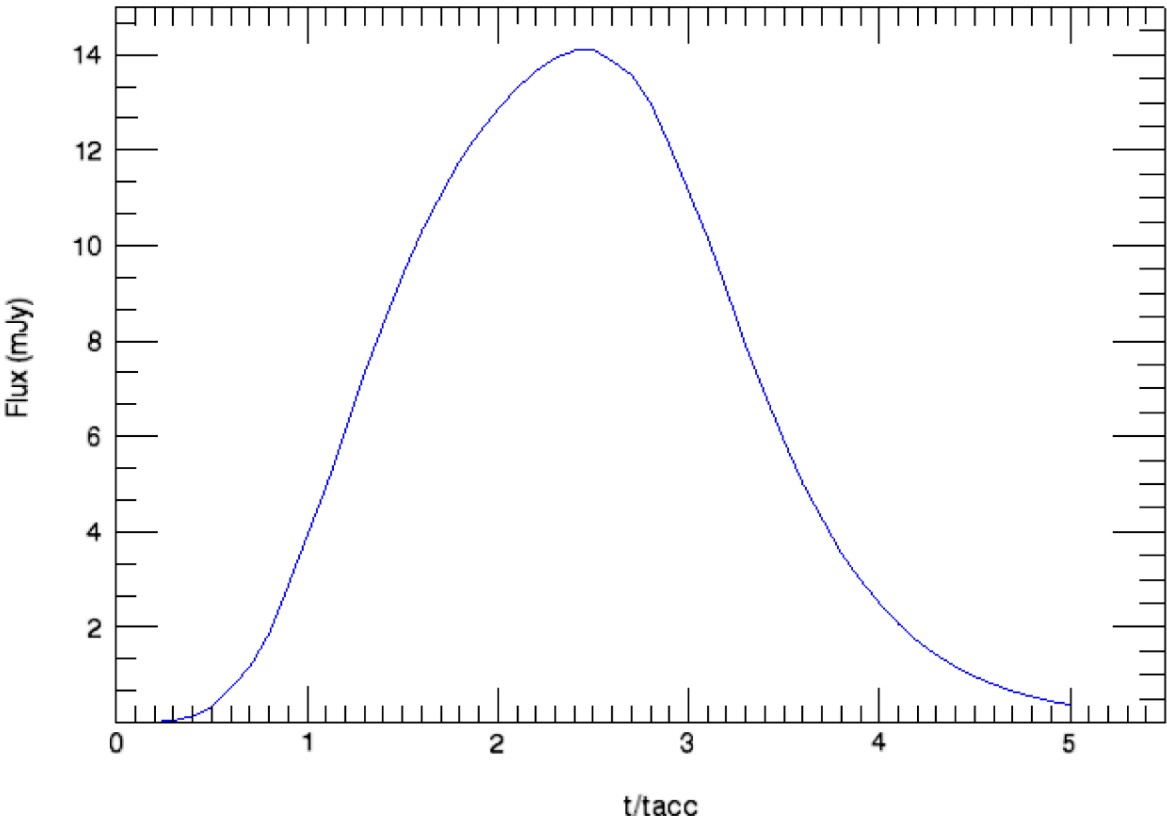

**Figure 2.** A typical pulse profile for shock encountering a density enhancement.

We re-calculated the pulse shape using the expected pulse profile for our frequency band and then compared these profiles to our micro-variability curves. Figure 3 shows a cartoon of a shock propagating from left to right down the jet and encountering multiple turbulent cells, which each emit a pulse of radiation in the shape of Figure 2. The KRM solutions constrain the shape of the pulse and give the amplitude and FWHM of an individual pulse, representing the emission from a single turbulent cell. We then have a linear combination of many pulses convolved together, plus a background emission component.

$$I(\nu, t) = I_{laminar} + \sum I_{cell(\nu, t)}$$

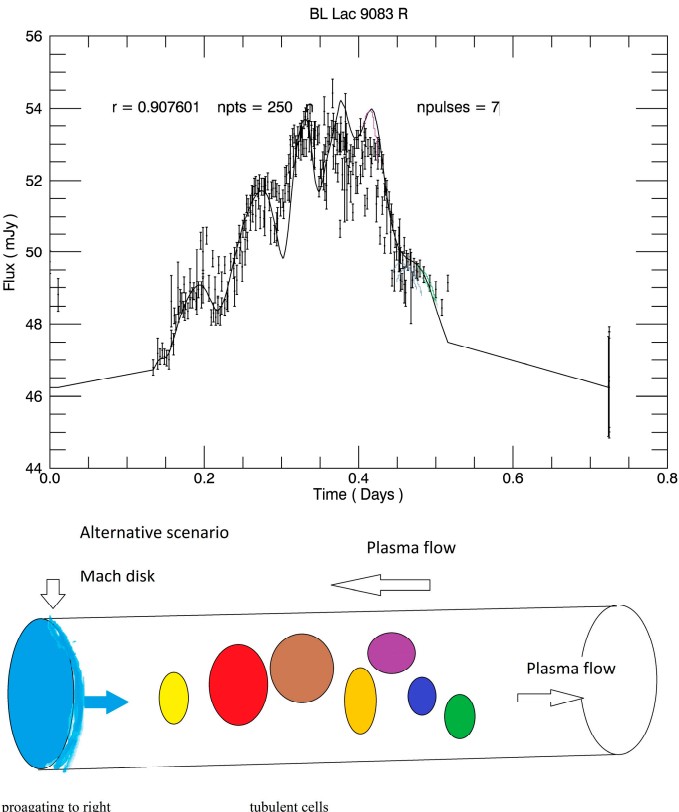

**Figure 3.** Diagram above shows a microvariability curve of BL Lac and demonstrates the general model attributes of a shock propagating from left to right through a turbulent jet (bottom illustration). The cartoon shows multicolored cells with the blue shock on the left side. As it propagates along the jet, it stimulates a pulse of radiation from each of the colored density enhancements. These pulses are convolved to produce the observed micro-variability curve shown. The alternative model is the plasma flowing from right to left encountering a Mach disk [5].

*3.2. IV Results: Fitting the Individual Light Curves*

To fit each of the 41 BL Lac micro-variability curves, we first identify the number of distinct, well-sampled peaks, and then determine the amplitude above the linear background. We estimate the shock speed relative to the jet flow following [5] and assume the relative velocity as c/3. The choice of the shock speed relative to the jet is the conversion between the FWHM duration of the pulse and the actual physical size of the turbulent cell. Other authors have used 0.9 c for the relative velocity, so one could multiply our resulting cell sizes by 3 to obtain absolute cell sizes based on 0.9 c [10]. The Doppler boost factor of the jet is assumed to be 7.5. We identify individual pulses in the light curve and estimate the center time, full width at half max, and amplitude for each pulse. The pulses are convolved together and the FWHM, center time, and amplitudes of the individual pulses are adjusted while keeping the shape constant until we maximize the correlation coefficient. Obviously, we cannot always de-convolve all of the pulses in a given light curve because of sampling and photometric accuracy. We attempted to model only the light curves which show well sampled pulses. The number of pulses fit in many individual micro-variability curves give us an idea of the distribution of the sizes of the turbulent cells in the plasma (assuming a constant shock speed). The individual fitted light curves are shown in Appendix A. The title of the plot is the modified and truncated JD date of the observation, r is the Pearson correlation coefficient of the resulting model fit, Npts is the number of points in the light curve and Npulses is the number of pulses used to attain the fit. Table 1 gives the results of the model fitting of the 41 individual light curves. We fit a total of 323 individual pulses to the data. The first column is the modified Julian date (JD-2450000.00) of the observation. Column 2 is the number of individual points in the light curve. Column 3 is the linear Pearson

correlation coefficient of the fit, and column 4 the number of pulses fit for that particular light curve. Column 5 is the F-test significance of each fit given the correlation coefficient and the degrees of freedom (NPTS − (3* number of pulses)). The fits were all very significant.

**Table 1.** Fit result.

| Mod JD | Npts | r | # Pulses | F-Test |
|--------|------|---|----------|--------|
| 9081 | 141 | 0.949 | 5 | <0.0001 |
| 9082 | 209 | 0.9623 | 5 | <0.0001 |
| 9083 | 250 | 0.9076 | 7 | <0.0001 |
| 9084 | 161 | 0.941915 | 7 | <0.0001 |
| 9086 | 299 | 0.95053 | 8 | <0.0001 |
| 9090 | 150 | 0.902 | 6 | <0.0001 |
| 9091 | 130 | 0.966 | 5 | <0.0001 |
| 9096 | 200 | 0.937 | 6 | <0.0001 |
| 9097 | 142 | 0.852587 | 6 | <0.0001 |
| 9099 | 371 | 0.926 | 6 | <0.0001 |
| 9100 | 171 | 0.945 | 8 | <0.0001 |
| 9101 | 788 | 0.978 | 9 | <0.0001 |
| 9104 | 319 | 0.992 | 8 | <0.0001 |
| 9105 | 296 | 0.84 | 8 | <0.0001 |
| 9106 | 401 | 0.945768 | 10 | <0.0001 |
| 9107 | 481 | 0.946 | 12 | <0.0001 |
| 9108 | 361 | 0.981 | 8 | <0.0001 |
| 9109 | 289 | 0.974 | 12 | <0.0001 |
| 9110 | 287 | 0.964718 | 10 | <0.0001 |
| 9112 | 348 | 0.945862 | 10 | <0.0001 |
| 9113 | 111 | 0.952 | 9 | <0.0001 |
| 9116 | 294 | 0.982511 | 10 | <0.0001 |
| 9117 | 232 | 0.936984 | 11 | <0.0001 |
| 9119 | 390 | 0.86825 | 12 | <0.0001 |
| 9120 | 123 | 0.979 | 4 | <0.0001 |
| 9121 | 99 | 0.994588 | 8 | <0.0001 |
| 9122 | 138 | 0.975509 | 7 | <0.0001 |
| 9123 | 176 | 0.989076 | 8 | <0.0001 |
| 9124 | 227 | 0.952871 | 6 | <0.0001 |
| 9125 | 203 | 0.939568 | 7 | <0.0001 |
| 9129 | 400 | 0.971661 | 9 | <0.0001 |
| 9130 | 337 | 0.988759 | 10 | <0.0001 |
| 9131 | 355 | 0.989075 | 7 | <0.0001 |
| 9132 | 118 | 0.96131 | 5 | <0.0001 |
| 9133 | 108 | 0.961 | 7 | <0.0001 |
| 9134 | 116 | 0.969 | 6 | <0.0001 |
| 9135 | 135 | 0.96 | 6 | <0.0001 |
| 9136 | 187 | 0.935698 | 8 | <0.0001 |
| 9141 | 133 | 0.945698 | 8 | <0.0001 |
| 9170 | 146 | 0.872274 | 7 | <0.0001 |
| 9173 | 125 | 0.894287 | 7 | <0.0001 |
| **Averages** | **242** | **0.948 ± 0.04** | **7.884** | <0.0001 |

Table 1 demonstrates how well the model pulses fit the data. The first column lists the modified and truncated modified Julian date of the dataset (JD-2450000), column 2 gives the number of data points in the micro-variability curve for that night, column 3 is the resulting correlation coefficient of the pulse fit to the data, and the fourth column is the number of pulses. The F-test is the significance of the fit given the degrees of freedom and the correlation coefficient.

Table 2 summarizes the results of the fits to the entire data set. We give the cell size results for two cases. The first is the case where the relative velocity between the jet flow and the shock is 0.3c and the second is the relative velocity is 0.9c.

**Table 2.** Pulse FWHM results.

|  | amp | Width AU ($v_{rel}$ = 0.3 c) | Width AU ($v_{rel}$ = 0.9 c) |
|---|---|---|---|
| average | 37.22 | 1.24 | 3.732 |
| min | 0.016 | 0.150618 | 0.451 |
| max | 19.27 | 4.01647 | 12.049 |
| st.dev | 3.377 | 0.77 | 2.31 |

The average, minimum and maximum of the amplitudes and the widths of the pulses are identified in Table 2. Column 2 gives the amplitude above the background, column 3 the cell size in AU if the shock speed is 0.3 c, and column 4 the cell size if the shock speed is 0.9 c.

Simulations of turbulence in blazar jets usually start with an assumption of the cell size. Pollack et al. [10] assumed 20 cells across the jet, while [5] assumed over 1100 zones in the jet. Marscher [5] uses a radius of the cylindrical cells in his TEMZ model simulations of 412 AU, with as many of 19 cells across the jet.

Here, we associate each "pulse" in the micro-variability light curve with an individual turbulent cell encountering the shock in the turbulent jet. Given the plasma flow speed relative to the shock, we can estimate the turbulent cell size. The amplitude of the pulse is related to both the particle density in that cell (the number of particles radiating synchrotron) and the angle of the magnetic field relative to the observer (synchrotron radiation depends on the $\sin(\phi)$), where $\phi$ is the angle between the magnetic field and the particle velocity). The magnetic field orientation will be randomly oriented in adjacent cells in a turbulent plasma.

An independent analysis of other data from an August 2020 flare of BL Lac was published by [11]. They used the pulse model to determine the range of cell sizes (4.4–77 AU) and the smallest cell size as 4.4 AU, imposing a relative velocity of 0.9 c. Previously, Meng [12] found cell sizes using this model as small as 1.5 AU assuming a speed of 0.1 c. Marscher [13] used a bulk laminar velocity of the unshocked plasma of 0.99 c before it encountered the Mach disk or standing shock in his TEMZ model.

The extreme, apparently random shifts in optical polarization angle and intensity seen in blazar observations are consistent with this interpretation since most polarization observations would likely convolve many turbulent cells because of the long duration of the polarization measurement. Correlating polarization shifts with de-convolved pulses is an interesting test of the model [2]. Polarization observations of S5 0716 + 71 led [14] to suggest that the polarization behavior seen suggests that the observed "micro-variability originated in a small and local region of the jet"! The general polarimetric behavior of blazars as seen by the WEBT observers [15] led them to conclude that "the magnetic field must have a turbulent component". Accurate time-resolved polarization measurements should be able to de-couple the density from the magnetic field direction in the pulse amplitude [16]. Although there were some polarization measurements made in conjunction with this current observation, the temporal density of the polarimetry was not sufficient to resolve the polarization of the individual pulses.

## 4. Conclusions

We have applied our turbulent cell model of blazar micro-variability to an extended set of optical observations of BL Lac gathered by the WEBT consortium during a 2020 WEBT campaign. On 41 of the nights of the BL Lac campaign, BL Lac showed substantial micro-variability. We applied our model to each of those nights independently to determine the distribution of cell sizes and amplitudes in the BL Lac jet. The pulse shapes were calculated using the KRM model [2,6] and the average correlation coefficient of the fits was 0.948, while the best fit yielded a correlation coefficient of r = 0.995. We associate each individual pulse with a turbulent cell in the jet plasma, which is either overtaken by a moving shock or encounters a Mach disk or standing shock at a velocity of 0.3 c. These results can be used to inform the boundary conditions for the numerical simulation of turbulence in blazar jets. The model also makes predictions about polarization variability, but the polarization data published in this campaign were not temporally concentrated enough to test these predictions.

**Author Contributions:** J.R.W. was program leader and principle author. I.P.S. contributed to fitting the model to the light curves. All authors have read and agreed to the published version of the manuscript.

**Funding:** This research received no external funding.

**Data Availability Statement:** The data is the property of the WEBT consortium.

**Acknowledgments:** The authors would like to acknowledge support from NASA CRE2DO grant at FIU in carrying out this research. The authors wish to thank Claudia Raiteri and Svetlana Jorstad and the observers in the WEBT consortium for access to the raw data used in this study. Based on data taken and assembled by the WEBT collaboration and stored in the WEBT archive at the Osservatorio Astrofisico di Torino—INAF (https://www.oato.inaf.it/blazars/webt/ (accessed on 24 October 2023)).

**Conflicts of Interest:** The authors declare no conflict of interest.

## Appendix A

Individual light curves. The title is the JD date of the observation, the r is the Pearson correlation coefficient for the resulting fit, npts is the number of points in the light curve and npulses is the number of pulses for the fit. The solid line is the model fit.

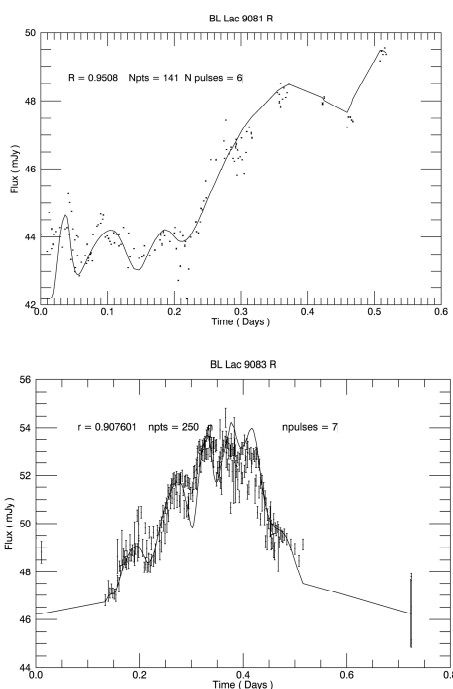

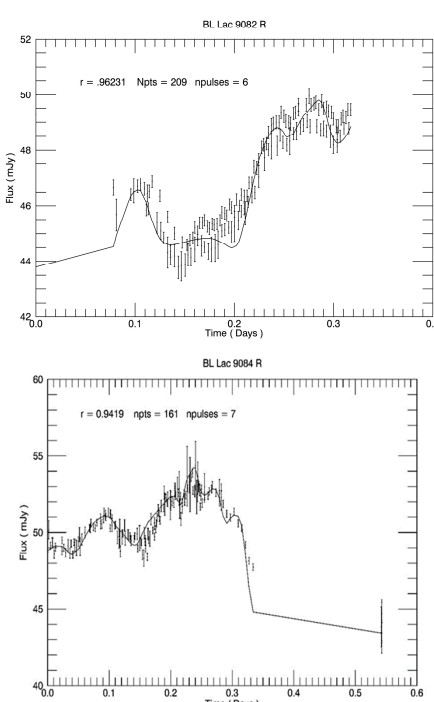

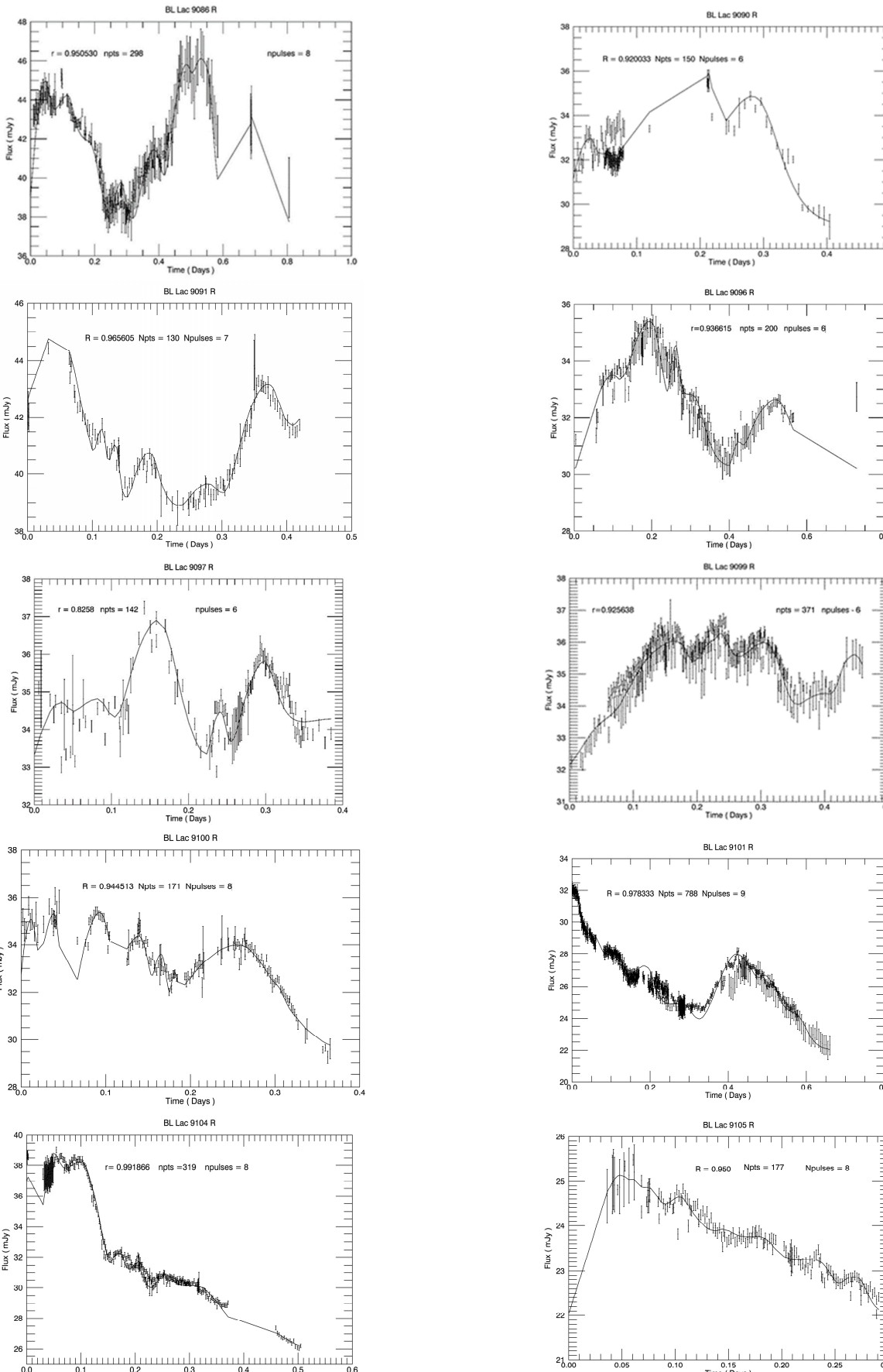

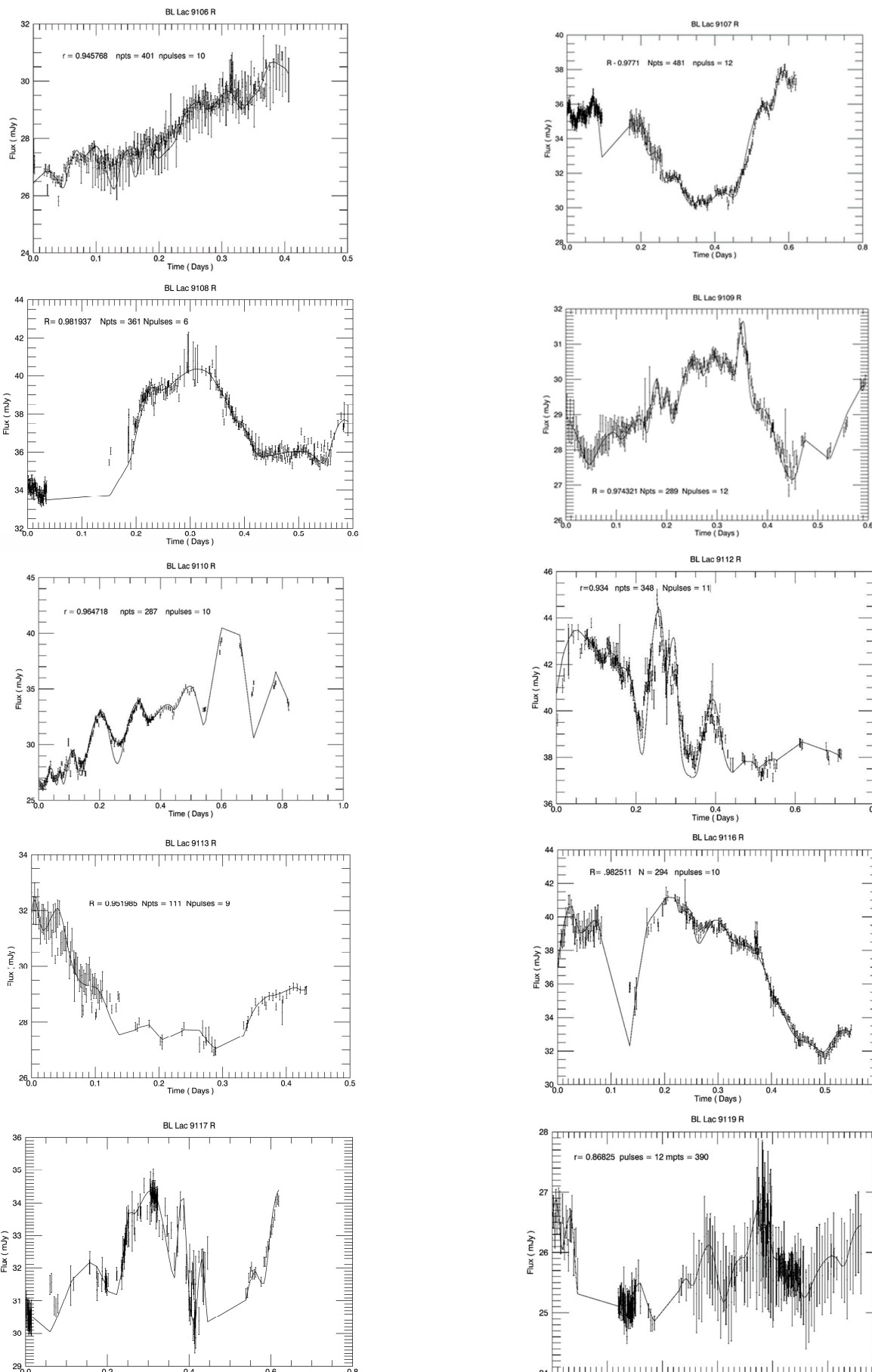

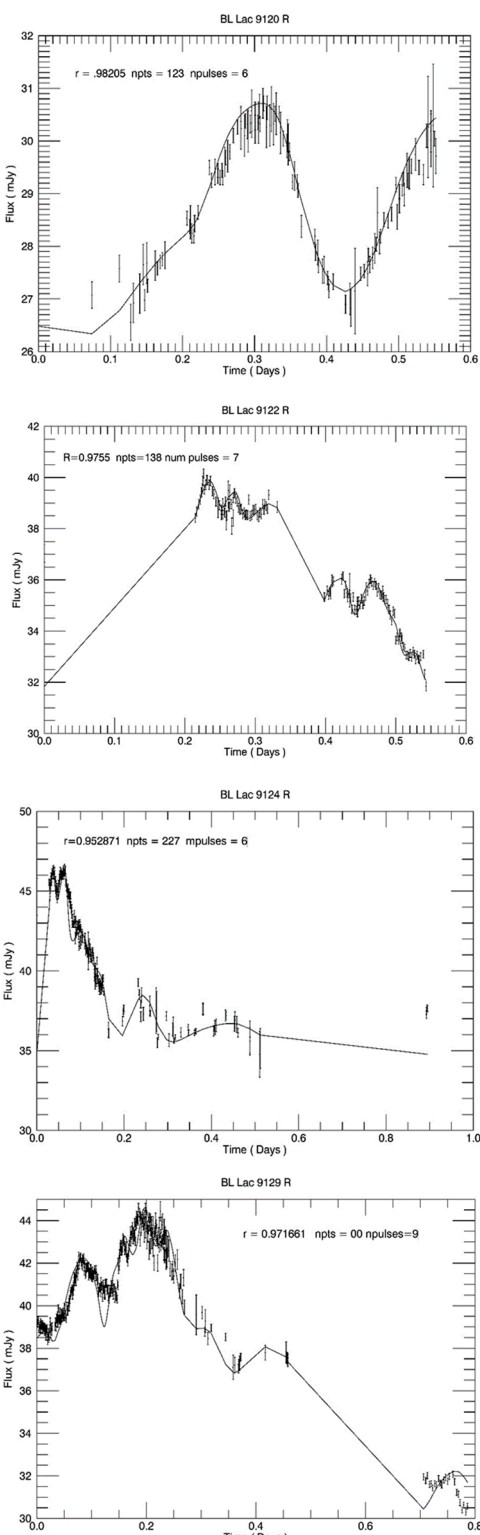

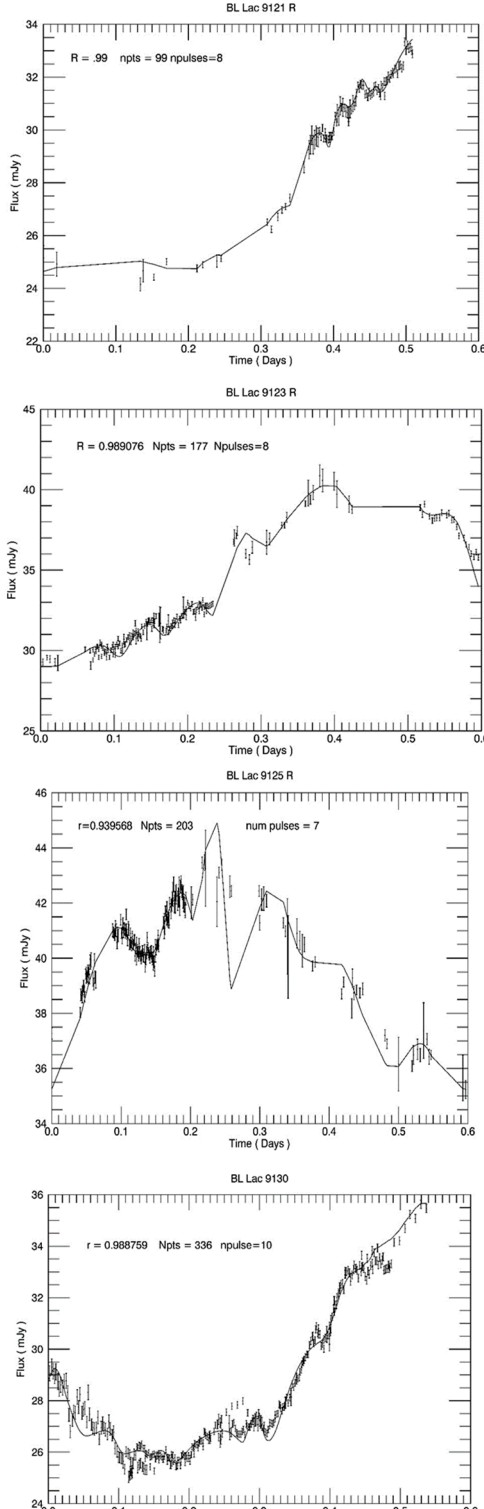

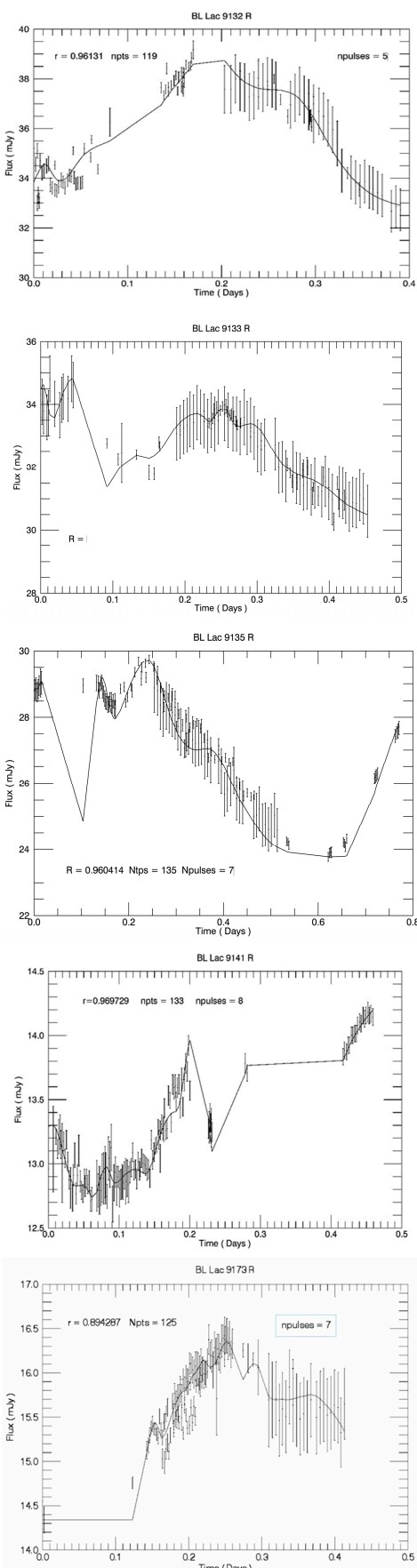

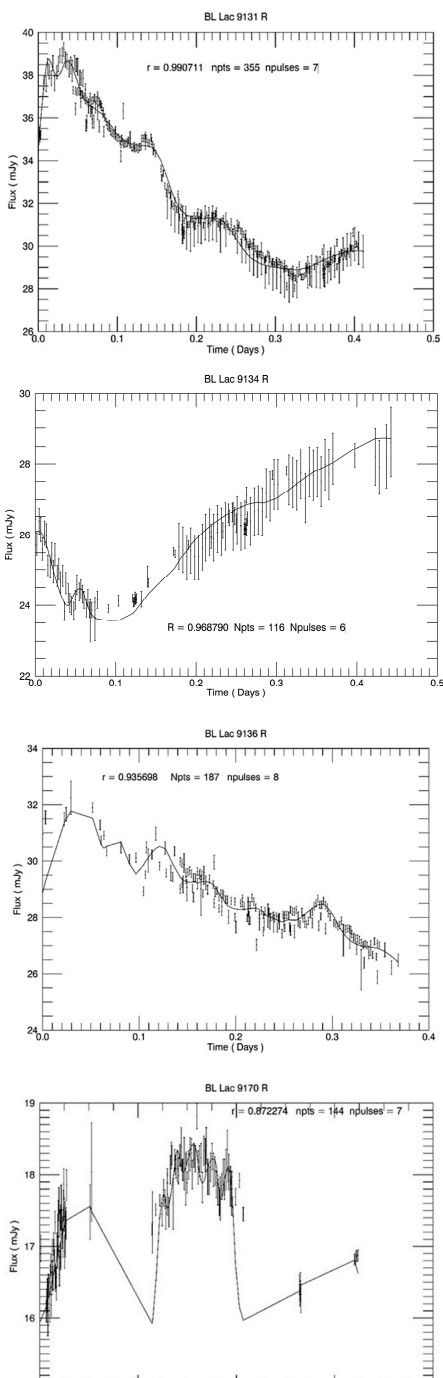

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
