# Peer review of "The Structure of Micro-Variability in the WEBT BL Lacertae Observation"

_galaxies, doi:10.3390/galaxies11060108_

Round 1
Reviewer 1 Report
Comments and Suggestions for Authors
The authors apply a pulse model that they published in an earlier paper to the light curve of the blazar BL Lacertae. The data come from an observing campaign in 2020 by the Whole Earth Blazar Telescope, with the addition of data from the Florida International University. The idea is that microvariability can be explained by the superposition of the emission contributions from turbulent cells inside the jet, where particles are accelerated by the passage of a shock wave, releasing pulses of synchrotron radiation. The model fit allows the authors to derive the number of the emitting cells contributing to each segment of light curve considered, and their size.
The paper is very interesting and deserves publication, but the text needs some revision.
In particular, I think that some more details on the model should be given in Section 3, so that the reader is not obliged to go back to the previous paper Webb et al. (2021).
In the following I list the points I noticed.
Line 7, Please specify the acronym WEBT and possibly give its web address.
Line 8 "in early 2020" delete "early" because the extension of the WEBT data reaches the end of 2020.
Line 8 Reference 1 should be within squared brackets.
Line 29 Some text seems to be missing: "with the width"?
Line 41 I think 0.54% is a typo, it cannot be the duty cycle of BL Lacertae!
Line 41 Insert "in 2020", i.e.: "In 2020 the WEBT collaboration observed BL Lacertae over a period of 231 nights". The clause is necessary because the WEBT has been observing BL Lacertae since the Collaboration birth in 1997 and the number of WEBT observing nights on BL Lacertae is several thousands.
Line 44 Please specify what FIU stands for.
Line 45 Typo: The abscissa in Figure 1 says "Modified JD", which is "Julian date minus 2450000.5", and does not correspond to the "Julian date minus 24580000" stated in the text, even when corrected for the typo. See also Line 119, where there is half a day difference with respect to the correct definition of MJD. Only at line 127 it comes out that the authors use a "modified and truncated Julian date". Please clarify since the beginning.
Line 50 Here the night selection criterion is "over 100 individual images", while at Line 10 it was "over 150 individual observations"; from table 1 one infers that "over 100" is the correct statement.
Line 51 "in" is missing: "are listed in table 1..."
Line 72-74 Verb is missing, maybe "The model predicts a shock wave..."?
Lines 86-87 Here some more details should be given.
Line 87 Figure 2 should be Figure 3
Line 89 figure 1 should be figure 2
Line 98 In the caption Figure 1 should be Figure 3. The lower, coloured part of the picture should be described, also linking it to the upper part.
Line 123 Delete the opening bracket just before the number 3.
Line 126 Typo: "fits" should be "fit".
Lines 126-131 This is a repetition of the content in lines 118-123, so it can be deleted. Moreover, the degrees of freedom is not "the number of pulses times 3", but it is the number of points minus the number of pulses times 3, as stated in lines 122-123.
Lines 136 and Table 2 Please write 0.3c and 0.9c
Line 138 Typo: "starts" should be "start".
Line 148 "orientation" can be deleted.
Line 149 "An indepedent..." I would go to a new line and I would specify that the papers that are cited below refer to BL Lacertae too.
Line 161 I would not use "recent" referring to results published in 2008...
Line 174 The reference for the KRM model is number 6
Appendix: information on r, number of points and number of pulses is missing in the second panel of the figure; moreover, such information is not given in a uniform form in the various panels
Author Response
In particular, I think that some more details on the model should begiven in Section 3, so that the reader is not obliged to go back to the previous paper Webb et al. (2021). I expanded a bit, but the editors requested I shorten the explanation from the original manuscript as it was repeatative of a previous paper.
In the following I list the points I noticed.
Line 7, Please specify the acronym WEBT and possibly give its web address. Done
Line 8 "in early 2020" delete "early" because the extension of the WEBT data reaches the end of 2020. Done
Line 8 Reference 1 should be within squared brackets. Done
Line 29 Some text seems to be missing: "with the width"? Fixed
Line 41 I think 0.54% is a typo, it cannot be the duty cycle of BLLacertae! The actual number is 54.73. Remember this is not just variability, but our new definition of microvariability. This is determined from over 53 microvariability observations over abpout 30 years of monitoring. Nights where it shows linear variations are not included.
Line 41 Insert "in 2020", i.e.: "In 2020 the WEBT collaboration observed BL Lacertae over a period of 231 nights". The clause is necessary because the WEBT has been observing BL Lacertae since the Collaboration birth in 1997 and the number of WEBT observing nights on BL Lacertae is several thousands. This is the period of time where the data is more or less continuous and used in this study. Added text in orange. The WEBT collaboration observed BL Lacertae over a period of 231 nights during the 2020 campaign.
Line 44 Please specify what FIU stands for. Added “Florida International University”
Line 45 Typo: The abscissa in Figure 1 says "Modified JD", which is "Julian date minus 2450000.5", and does not correspond to the "Julian date minus 24580000" stated in the text, even whencorrected for the typo. See also Line 119, where there is half a day difference with respect to the correct definition of MJD. Only at line127 it comes out that the authors use a "modified and truncatedJulian date". Please clarify since the beginning. The abscissa is modified Julian date minus 2450000.5
Line 50 Here the night selection criterion is "over 100 individual images", while at Line 10 it was "over 150 individual observations";from table 1 one infers that "over 100" is the correct statement. Fixed
Line 51 "in" is missing: "are listed in table 1..." fixed
Line 72-74 Verb is missing, maybe "The model predicts a shockwave..."? The model predicts that a shock wave encountering a cylindrical density enhancement in the plasma accelerates electrons which then cool by emitting synchrotron radiation.
Lines 86-87 Here some more details should be given. The flux rise and decline of the model pulse closely matched the observed Blazar micro-variability pulses.
Line 87 Figure 2 should be Figure 3 fixed, thanks!
Line 89 figure 1 should be figure 2fixed
Line 98 In the caption Figure 1 should be Figure 3. The lower,coloured part of the picture should be described, also linking it tothe upper part. Figure 3. Diagram above shows the light curve of ON 231 and demonstrates the general model attributes of a shock propagating from left to right through a turbulent jet (bottom illustration). The cartoon shows multicolored cells with the blue shock on the left side. As it propagates along the jet it stimulates a pulse of radiation from each of the colored density enhancements. These pulses are convolved to produce the observed micro-variability curve shown. That was the original object that I got the idea of the model from. However I made a new figure for this paper using a BL Lac light curve as the reviewer suggested. Remember it is just a cartoon to show the concept.
Line 123 Delete the opening bracket just before the number 3. Added closing bracket after pulses) so it becomes (NPTS-(3*number of pulses)).
Line 126 Typo: "fits" should be "fit". Fixed
Lines 126-131 This is a repetition of the content in lines 118-123,so it can be deleted. Moreover, the degrees of freedom is not "thenumber of pulses times 3", but it is the number of points minus the fixed
Reviewer 2 Report
Comments and Suggestions for Authors
This article studied the micro-variability in the lightcurve of BL Lacertae measured by WEBT and derived the properties of turbulent plasma in the jet based on the turbulent cell model. In general, I found the article interesting and may provide valuable addition to the on-going discussion of the energy dissipation mechanism in blazars. I have a few comments/suggestion to the article, and would recommend it for publication after the authors address them.
1. the third figure of this article, which is mistakenly labelled as Figure 1 at page 4, was actually shown already in a published paper (reference 2) at the same journal Galaxy. Although the first author of that paper is the same as the submitted article, I think it is inappropriate to use the same figure in this article without pointing out the original publication. In addition, this figure is about the lightcurve of another blazar ON 231, while this article focuses on a different blazar, BL Lacertae. It seems to me confusing to show the figure in this article. Can the author make a similar figure but for BL Lacertae instead?
2. when the authors mention the correlation coefficient several times in the article, they do not introduce which kind of the correlation coefficient they use here. The relevant information need be provided.
3. the figure in Appendix. Please clarify the meanings of symbols and curves in these panels. Also, I found that the image resolution of the top two panels are worse than others. It'd better to fix them.
4. I am wondering that what would the author's model predict for the lightcurve in the gamma-ray band? Fermi-LAT can provide a long-term monitor of the lightcurve for the blazar. Is it consistent with the WEBT lightcurve and useful to test the model?
Author Response
This article studied the micro-variability in the light curve of BL Lacertae measured by WEBT and derived the properties of turbulent plasma in the jet based on the turbulent cell model. In general, I found the article interesting and may provide valuable addition to the on-going discussion of the energy dissipation mechanism in blazars. I have a few comments/suggestion to the article, and would recommend it for publication after the authors address them.
the third figure of this article, which is mistakenly labelled as Figure 1 at page 4,(Fixed) was actually shown already in a published paper (reference 2) at the same journal Galaxy. Although the first author of that paper is the same as the submitted article, I think it is inappropriate to use the same figure in this article without pointing out the original publication. In addition, this figure is about the light curve of another blazar ON 231, while this article focuses on a different blazar, BL Lacertae. It seems to me confusing to show the figure in this article. Can the author make a similar figure but for BL Lacertae instead?
- This figure is included as it is representative of the light curve with well separated pulses that led to this model. It was realized by looking at this particular light curve that the micro variability could be de convolved into individual pulses. It is also important to know that this shape works on all objects in addition to BL Lac!
- when the authors mention the correlation coefficient several times in the article, they do not introduce which kind of the correlation coefficient they use here. The relevant information need be provided. I added the correlation coefficient is a linear Pearson correlation coefficient.
- the figure in Appendix. Please clarify the meanings of symbols and curves in these panels. Also, I found that the image resolution of the top two panels are worse than others. It'd better to fix them. added text: The title of the plot is the Modified and truncated JD date of the observation, the r is the Pearson correlation coefficient of the resulting model fit, Npts is the number of points in the light curve and Npulses is the number of pulses used to attain the fit.
- I am wondering that what would the author's model predict for the light curve in the gamma-ray band? Fermi-LAT can provide along-term monitor of the lightcurve for the blazar. Is it consistent with the WEBT lightcurve and useful to test the model? This model is only for synchrotron emission that does not extend to the gamma-ray region of the spectrum. However Alan Marschers multi-zone turbulent jet model does include gamma-ray emission is is very successful in modelling the multifrequency aspects assuming a turbulent plasma.
Round 2
Reviewer 2 Report
Comments and Suggestions for Authors
I think this version is suitable for publication.